

# Comparison of the efficacy of 12 interventions in the treatment of diabetic foot ulcers: a network meta-analysis

Xuyang Hu[1,2,*], Huixin Meng[2,*], Jiaru Liang[2], Hang An[2], Jiaqi Zhou[1], Yuling Gao[1], Chong You[2], Zhenni Zhang[1], Xiaoyang Gong[1] and Yong Liu[1,2]

[1] Department of Rehabilitation Medicine, The First Affiliated Hospital of Dalian Medical University, Dalian, China

[2] Institute (College) of Integrative Medicine, Dalian Medical University, Dalian, China

[*] These authors contributed equally to this work.

Corresponding authors
Xiaoyang Gong,
rowlandgxy@163.com
Yong Liu, fuwa52008@126.com

## ABSTRACT

**Objective**. This study aimed to comprehensively compare the efficacy of 12 interventions for diabetic foot ulcer (DFU) using a network meta-analysis (NMA).

**Methods**. The NMA was conducted by PRISMA guidelines, and the protocol was registered in PROSPERO (CRD42023461811). PubMed, Web of Science, Cochrane Library, and Embase databases were systematically searched from inception to September 2023. Randomized controlled trials (RCTs) enrolling patients with DFU were included if they compared epidermal growth factor (EGF), platelet-derived growth factor (PDGF), platelet-rich plasma (PRP), stem cells (SC), low-frequency ultrasound (LFU), negative pressure wound therapy (NPWT), low-level laser therapy (LLLT), electric stimulation (ES), extracorporeal shockwave therapy (ESWT), amniotic membrane therapy (AMT), hyperbaric oxygen therapy (HBOT), and topical oxygen therapy (TOT) against standard of care (SOC) or placebo. The primary endpoint assessed was the wound healing rate. Secondary endpoints comprised wound healing time, percentage area reduction (PAR), and amputation rate. The surface under the cumulative ranking curve (SUCRA) was calculated to rank the efficacy of interventions.

**Results**. A total of 99 RCTs involving 7,356 patients were included. Among the 12 interventions analyzed, only LFU (OR = 2.20; 95% CI [0.99–4.91]) and ES (OR = 1.88; 95% CI [0.87–4.05]) did not demonstrate statistically significant improvements in ulcer healing rate compared with SOC. Based on SUCRA rankings, SC (SUCRA = 89.7%; OR = 5.71; 95% CI [2.64–12.34]) and AMT (SUCRA = 89.2%; OR = 5.11; 95% CI [3.12–8.37]) ranked highest in promoting ulcer healing, while LFU (29.4%) and SOC (10.4%) ranked lowest. Regarding wound healing time, AMT (MD = −26.91 days; 95% CI [−44.27 to −9.55]), PRP (MD = −21.65 days; 95% CI [−33.61 to −9.69]), and NPWT (MD = −16.79 days; 95% CI [−31.12 to −2.26]) significantly reduced healing durations compared to SOC. SUCRA rankings indicated that AMT (84.7%) and PRP (74.6%) ranked highest, while LFU (29.4%) and SOC (10.4%) remained lowest. Concerning PAR, LLLT (MD = 34.27; 95% CI [17.35–51.20]) and ESWT (MD = 27.50; 95% CI [11.00–44.00]) showed significant improvements over SOC, with LLLT (SUCRA = 93.9%) and ESWT (SUCRA = 84.0%) ranking highest, while SOC (21.0%) and TOT (18.3%) ranked lowest. For amputation rate, SC (OR = 0.12; 95% CI [0.03–0.55]) and HBOT (OR = 0.35; 95% CI [0.16–0.78]) significantly lowered the

risk compared to SOC, with SUCRA rankings placing SC (79.9%) and PRP (73.2%) as most effective, while NPWT (26.4%) and SOC (9.9%) were least effective.

**Conclusions**. SC and AMT emerged as highly effective, demonstrating superior efficacy in improving healing rate compared to PDGF, ES, and HBOT. AMT also showed significant effects in shortening ulcer healing time. LLLT exhibited considerable effectiveness in reducing ulcer areas, and SC therapy was associated with reduced amputation rate.

# INTRODUCTION

Diabetes has become one of the most serious and common diseases in the world. In 2021, diabetes affected 10.5% of the global population aged 20 to 79 years, corresponding to approximately 536.6 million individuals. This prevalence is anticipated to increase to 12.2%, impacting around 783.2 million people by the year 2045 (*Sun et al., 2022a*). Diabetic foot ulcer (DFU) is a serious complication of diabetes, characterized by skin breakdown in the feet, involving at least the epidermis and part of the dermis, usually accompanied by lower limb neuropathy and/or peripheral arterial disease (*Van Netten et al., 2024*). Epidemiological studies suggest that 19–34% of diabetic patients will suffer from DFU, about 60% of DFU patients will develop infection, about 20% will require lower limb amputation, and 10% will die within the first year of diagnosis (*McDermott et al., 2023*). DFU imposes a substantial economic burden on healthcare systems and individual expenses, amounting to US$9 to 13 billion annually, which is two to three times higher than that for non-DFU patients (*Rice et al., 2014*). The development of DFU is mainly associated with four aspects: ischemia caused by peripheral arterial disease, leading to tissue necrosis due to insufficient blood supply; peripheral neuropathy leads to sensory, motor and secretory dysfunction in the lower limb skin, resulting in loss of protection and delayed healing; chronic inflammation caused by bacterial infection also delays healing and cell dysfunction hinders wound repair (*Deng et al., 2023a*). The normal wound healing process includes four phases: hemostasis, inflammation, proliferation, and remodeling (*Mamun et al., 2024*). However, in diabetic wounds, these programmed phases are disrupted, preventing normal wound healing. This complex pathological process brings unique challenges to the management of DFU, necessitating advanced therapeutic strategies to enhance patient outcomes.

Currently, the clinical management of DFU primarily relies on standard of care (SOC), including local debridement, wound dressing, offloading, revascularization, and infection control (*Schaper et al., 2024*). These approaches greatly relieve symptoms and promote wound healing, constituting a fundamental clinical strategy. Over the past decades, a variety of advanced interventions have been explored to improve DFU healing. Such interventions encompass a range of therapies: epidermal growth factor (EGF), platelet-derived growth
factor (PDGF), platelet-rich plasma (PRP), stem cells (SC), low-frequency ultrasound (LFU), negative pressure wound therapy (NPWT), low-level laser therapy (LLLT), electric stimulation (ES), extracorporeal shockwave therapy (ESWT), amniotic membrane therapy (AMT), hyperbaric oxygen therapy (HBOT), and topical oxygen therapy (TOT) (*Armstrong et al., 2023*; *Ganesan & Orgill, 2024*). These therapies have shown promising potential. Several randomized controlled trials (RCTs) have demonstrated promising efficacy for these interventions. For instance, AMT, SC therapy, and LLLT have revealed improved wound healing rate, shortened healing time, and reduced wound area (*DiDomenico et al., 2018*; *Haze et al., 2022*; *Uzun et al., 2021*). Additionally, multiple meta-analyses have confirmed that these treatments are superior in terms of healing rate, area reduction, and amputation prevention. A meta-analysis of nine RCTs involving 720 participants confirmed that EGF significantly improved the healing rate compared to SOC (OR: 2.79; 95% CI [1.99–3.99]) and significantly shortened healing time (MD: −14.10 days; 95% CI [−18.03 to −0.16]) (*Zhao et al., 2020*). Similarly, a meta-analysis of 22 RCTs (1,559 patients) revealed that PRP significantly enhanced the healing rate (RR: 1.42; 95% CI [1.30–1.56], $P < 0.001$), and reduced healing time (MD: −3.13 days; 95% CI [−5.86 to −0.39], $P < 0.001$) compared with SOC (*Deng et al., 2023b*). In addition, SC therapy, LLLT, ES, NPWT, AMT, HBOT, and TOT have been associated with significant enhancements in healing rate, along with a decrease in ulcer size and a lower risk of amputation across multiple RCTs (*Chen et al., 2021*; *Chen et al., 2020*; *Huang et al., 2021*; *Huang et al., 2023*; *Mohammed et al., 2022*; *Sun et al., 2022b*; *Tao & Yuan, 2024*).

Despite promising findings from individual interventions, existing guidelines highlight variations in the effectiveness of different treatments for DFU (*Chen et al., 2024*). Most published meta-analyses in the field of DFU have adopted traditional pairwise comparison methods. However, with the increasing number of clinical interventions for DFU, pairwise comparisons alone are insufficient to meet the needs of comprehensive evaluations. To date, two network meta-analyses (NMAs) have compared more than three interventions (*OuYang et al., 2024*; *Qian et al., 2024*). However, the interventions included were not comprehensive, omitting several commonly used clinical treatments. Therefore, this study aims to employ NMA, which enables indirect comparisons through a common control group, providing a more comprehensive evaluation of relative effectiveness (*Salanti et al., 2008*), to compare the effectiveness of 12 interventions for DFU.

## METHODS

This NMA adhered to the Preferred Reporting Items for Systematic Reviews and Meta-Analyses (PRISMA) guidelines. To ensure greater transparency and reproducibility of the research, the study protocol was prospectively registered in the PROSPERO database (CRD42023461811).

### Data sources and search strategies

Two authors (HXY and MHX) independently searched PubMed, Web of Science, Cochrane Library, and Embase from inception to September 2023, utilizing both Medical Subject Headings (MeSH) and free-text terms. As NMA requires data from at least three RCTs, we

identified 13 additional intervention types beyond the SOC through preliminary searches. The complete search strategies for each database, including detailed search terms, are provided in Supplemental Information 2.

## Study selection

We included RCTs evaluating 12 interventions for treating patients diagnosed with DFU, including EGF, PDGF, PRP, SC, LFU, NPWT, LLLT, ES, ESWT, AMT, HBOT, and TOT. Studies were eligible if the control group received SOC or a placebo. Studies comparing two or more interventions were included if they met the other criteria. Included studies were required to report the outcome of wound healing rate, wound healing time, percentage area reduction (PAR), or amputation rate, including both major and minor amputations. Eligible studies needed to clearly define and report these outcome measures and be published in English.

We excluded studies involving patients without DFU, non-RCTs, and studies that lacked clearly defined outcome measures. Studies with fewer than 10 participants or a total study duration shorter than 4 weeks (from intervention initiation to final assessment) were also excluded. Of the 13 interventions identified during the preliminary search, 12 were included in the final analysis because the number of eligible RCTs investigating ozone therapy was inadequate to satisfy the inclusion criteria.

## Data extraction and quality assessment

Two authors, MHX and LJR, independently extracted data from the included RCTs. Any discrepancies were reconciled through consensus with the involvement of a third reviewer, HXY. The extracted data included study characteristics (author, year, country, and sample size) and population demographics (age, gender, ulcer grade, and treatment duration). Additionally, details of interventions, control measures, and clinical outcomes were collected. Two authors, YC and AH, evaluated the risk of bias using the Cochrane tool, focusing on factors such as random sequence generation, allocation concealment, and blinding. In instances of disagreement, consensus was achieved through discussion and subsequently adjudicated by a third author, HXY.

## Data analysis

We conducted a frequentist network meta-analysis using a random-effects model to simultaneously estimate the relative effects of all interventions by combining both direct and indirect evidence (*Higgins et al., 2012*). To ensure network connectivity and coherent synthesis, trials were aggregated into 12 intervention nodes according to their principal therapeutic modality (*e.g.*, growth-factor, physical, or cellular therapies). Analyses were conducted at the node level, assuming that treatments within each node share similar mechanisms of action and clinical intent, consistent with the transitivity assumption in NMA (*Cornell, 2015*). We assessed network consistency and overall heterogeneity in all treatment comparisons. Global inconsistency tests and the node-splitting method were used to evaluate disparities between direct and indirect evidence, with a $p$-value $< 0.05$ indicating significant heterogeneity or inconsistency. All statistical analyses were performed using STATA 16.0. Interventions were ranked based on their effect sizes by calculating

Surface Under the Cumulative Ranking (SUCRA) values. We explicitly defined the primary and secondary outcomes to assess the effectiveness of treatments on DFU healing. The primary endpoint assessed was the wound healing rate. Secondary endpoints comprised wound healing time, PAR, and amputation rate. Dichotomous variables were reported as odds ratios (OR) with 95% confidence intervals (CI), whereas continuous variables were expressed as mean differences (MD) with 95% CI. We conducted subgroup analyses to evaluate whether treatment effects varied based on study duration. Separate network meta-analyses were conducted for each outcome, stratified by follow-up duration ($\leq$12 weeks *vs.* >12 weeks).

## RESULTS

### Study characteristics

During our systematic search, we identified 1,981 references, of which 669 were duplicates, leaving 1,312 for detailed review. After the initial assessment of titles and abstracts, 879 articles were removed. Subsequently, 433 articles underwent full-text review for eligibility, and 319 studies were excluded based on predefined criteria, including non-DFU populations, lack of randomization, unavailable full-text data, and the use of combined interventions. An additional 15 studies were excluded based on insufficient duration and small sample sizes. Ultimately, 99 studies met the eligibility criteria and were included in the final analysis. The PRISMA flowchart (Fig. 1) illustrates the detailed selection process.

The included RCTs were primarily conducted in the USA, India, and China, with publication years ranging from 1992 to 2023. A total of 7,356 participants were enrolled, with sample sizes ranging from eight to 382 and participant ages spanning from 32 to 72 years. Detailed study characteristics are summarized in Table 1. These RCTs assessed a variety of treatments, including 10 comparing EGF with SOC (*Afshari et al., 2005*; *Fernández-Montequín et al., 2009*; *Gomez-Villa et al., 2014*; *Oliveira et al., 2021*; *Park et al., 2018*; *Singla et al., 2014*; *Tsang et al., 2003*; *Viswanathan, Juttada & Babu, 2020*; *Xu et al., 2018*; *Zhang et al., 2021a*), nine studies on PDGF (*Agrawal et al., 2009*; *Bhansali et al., 2009*; *Blume et al., 2011*; *Jaiswal et al., 2010*; *Khandelwal et al., 2013*; *Ma et al., 2015*; *Samuel et al., 2016*; *Steed, 1995*; *Wieman, Smiell & Su, 1998*), 11 on PRP (*Ahmed et al., 2017*; *Driver et al., 2006*; *Elsaid et al., 2020*; *Gowsick et al., 2023*; *Gupta et al., 2021*; *Hossam et al., 2022*; *Li et al., 2015*; *Malekpour Alamdari et al., 2021b*; *Orban et al., 2022*; *Singh et al., 2018*; *Xie et al., 2020*), eight on SC Therapy (*Lu et al., 2008*; *Han, Kim & Kim, 2010*; *Huang et al., 2005*; *Jain et al., 2011*; *Lu et al., 2011*; *Mohammadzadeh et al., 2013*; *Ozturk et al., 2012*; *Uzun et al., 2021*), five on LFU (*Abd El Fattah et al., 2018*; *Ennis et al., 2005*; *Lázaro-Martínez et al., 2020*; *Michailidis et al., 2018*; *Rastogi, Bhansali & Ramachandran, 2019*), eight on NPWT (*Anjum et al., 2022*; *Blume et al., 2008*; *Karatepe et al., 2011*; *Malekpour Alamdari et al., 2021a*; *Maranna et al., 2021*; *Nain et al., 2011*; *Seidel, Lefering & DiaFu study, 2022*; *Seidel et al., 2020*), six on LLLT (*Haze et al., 2022*; *Kaviani et al., 2011*; *Minatel et al., 2009*; *De Alencar Fonseca Santos et al., 2018*; *Naidu et al., 2005*), three on ES Therapy (*Baker et al., 1997*; *Lundeberg, Eriksson & Malm, 1992*; *Peters et al., 2001*), eight on ESWT (*Jeppesen et al., 2016*; *Moretti et al., 2009*; *Nossair, Eid & Salama, 2013*; *Omar et al., 2014*; *Sandoval*
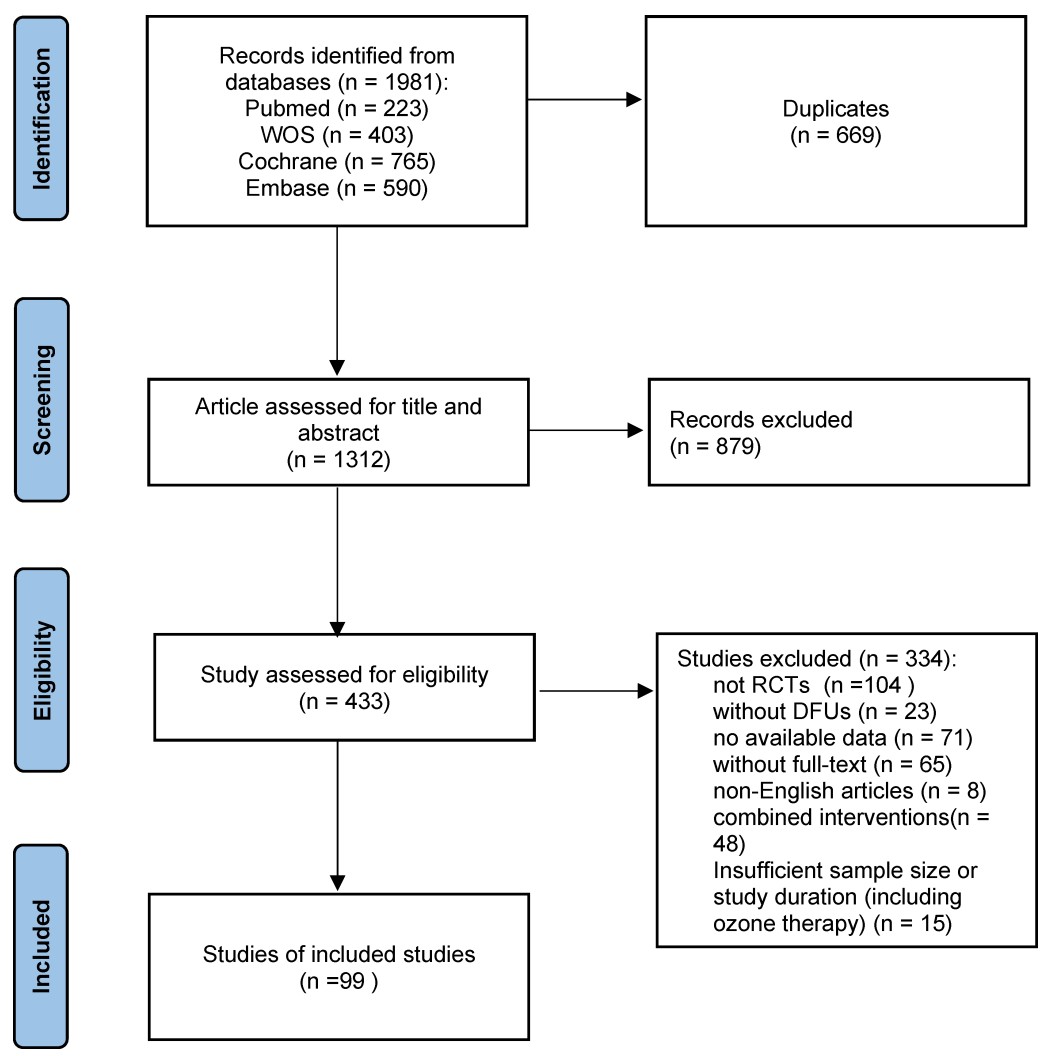

**Figure 1** **Flow chart of the study selection process.** A total of 1,981 records were identified from four databases. After removing duplicates and screening titles, abstracts, and full texts, 99 RCTs were included in the final analysis.

*Ortíz et al., 2014*; *Snyder et al., 2018*; *Vangaveti et al., 2023*; *Wang et al., 2009*; *Wang, Wu & Yang, 2011*), 12 on AMT (*DiDomenico et al., 2018*; *Game et al., 2021*; *Lavery et al., 2014*; *Mohajeri-Tehrani et al., 2016*; *Serena et al., 2020*; *Snyder et al., 2016*; *Tettelbach et al., 2019a*; *Tettelbach et al., 2019b*; *Thompson et al., 2019*; *Zelen et al., 2015*; *Zelen et al., 2013*; *Zelen et al., 2016*), 10 on HBOT (*Abidia et al., 2003*; *Chen et al., 2017*; *Duzgun et al., 2008*; *Faglia et al., 1996*; *Fedorko et al., 2016*; *Kessler et al., 2003*; *Kumar et al., 2020*; *Londahl et al., 2010*; *Salama et al., 2019*; *Santema et al., 2018*), and nine on TOT (*Al-Jalodi et al., 2022*; *Driver et al., 2017*; *Frykberg et al., 2020*; *He et al., 2021*; *Niederauer, Michalek & Armstrong, 2015*; *Niederauer, Michalek & Armstrong, 2017*; *Niederauer et al., 2018*; *Serena et al., 2021*; *Yu et al., 2016*). Figure 2 illustrates the treatment network for the various interventions.

**Table 1  Characteristics of the included studies.**

| Author year | Country | Size I/C | Gender distribution M/F | Comparisons | Mean age (years) | | Study duration | Treatment duration | Ulcer grade | Outcomes |
|---|---|---|---|---|---|---|---|---|---|---|
| Han 2010 | Korea | 26/26 | 29/23 | SC vs SOC | 66.5 | 68.4 | 8 weeks | 8 weeks | Wagner: 1–2 | ①② |
| Huang 2005 | China | 14/14 | 18/10 | SC vs SOC | 71.1 | 70.9 | 12 weeks | — | TEXAS: 1–2C, 1–3D | ①④ |
| Jain 2011 | India | 25/23 | 32/16 | SC vs SOC | 54 | 58 | 12 weeks | — | — | ① |
| Lu 2008 | China | 22/23 | 26/19 | SC vs SOC | 66.5 | 65.52 | 12 weeks | — | Fontaine grade IV | ①④ |
| Lu 2011 | China | 41/41 | — | SC vs SOC | 64 | 64 | 24 weeks | — | — | ①④ |
| Mohammadzadeh 2013 | Iran | 7/14 | — | SC vs SOC | 63.5 | 64.2 | 12 weeks | — | — | ④ |
| Ozturk 2012 | Turkey | 20/20 | 16/13 | SC vs SOC | 71.9 | 70.8 | 12 weeks | — | — | ① |
| Uzun 2021 | Turkey | 10/10 | 12/8 | SC vs SOC | 57.5 | 57.2 | 4 years | — | Wagner: 1–2 | ①②④ |
| Haze 2021 | Israel | 10/10 | 13/7 | LLLT vs SOC | 65 | 61 | 12 weeks | 12 weeks | TEXAS: IIA, IIIA/B, | ①③④ |
| Kaviani 2011 | Iran | 13/10 | 12/6 | LLLT vs SOC | 60.2 | 59.4 | 20 weeks | 20 weeks | Wagner: 1–2 | ①③④ |
| Minatel 2009 | USA | 12/11 | — | LLLT vs SOC | 66.3 | 63.4 | 12 weeks | 12 weeks | — | ①③ |
| Naidu 2005 | Malaysia | 8/8 | — | LLLT vs SOC | 40–50 | | 6 weeks | 6 weeks | with "Depth-Ischaemia Classification" grade 1 foot ulcers | ① |
| Sandoval Ortíz 2014 | Colombia | 9/10/9 | 12/16 | LLLT vs ES vs SOC | 59.3 | | 16 weeks | 16 weeks | Wagner: 1–2 | ① |
| de Alencar Fonseca Santos 2018 | Brazil | 9/9 | — | LLLT vs SOC | 53.11 | 48.33 | 4 weeks | 4 weeks | Noninfected lesion with II and III depth | ③ |
| Agrawal 2009 | India | 14/14 | 19/9 | PDGF vs SOC | 54.38 | 56.24 | 12 weeks | 12 weeks | Wagner: 1-4 | ① |
| Bhansali 2009 | India | 10/10 | 12/8 | PDGF vs SOC | 51.7 | 49.5 | 20 weeks | 20 weeks | Wagner: >2 | ①② |
| Blume 2011 | USA | 72/19 | 65/26 | PDGF vs SOC | 57.9 | 54.8 | 12 weeks | 12 weeks | Wagner: 1 | ① |
| Jaiswal 2010 | India | 25/25 | 42/8 | PDGF vs SOC | 56.20 | 49.92 | 10 weeks | 10 weeks | IAET stage III and IV | ① |
| Khandelwal 2013 | India | 20/20/20 | 32/28 | PDGF vs HBOT vs SOC | 43.35 (PDGF), 43.8 (HBOT) | 45 | 10 weeks | 10 weeks | IAET stage III and IV | ①② |
| Ma 2015 | USA | 23/23 | 46/0 | PDGF vs SOC | 59.3 | 60.1 | 40 weeks | 16 weeks | Wagner: 1 | ① |
| Samuel 2016 | India | 29 | 18/11 | PDGF vs SOC | 56.09 | — | 24 weeks | 24 weeks | Wagner: 1-3 | ① |
| Steed 1995 | USA | 61/57 | 89/29 | PDGF vs SOC | 63.2 | 58.3 | 20 weeks | until complete ulcer closure or up to 20 weeks | — | ① |
| Wieman 1998 | USA | 255/127 | 255/127 | PDGF vs SOC | 58 | | 32 weeks | 20 weeks | IAET stage III and IV | ① |
| Afshari 2005 | Iran | 30/20 | 27/23 | EGF vs SOC | 56.9 | 59.7 | 8 weeks | 4 weeks | Wagner: 1–5 | ①② |
| Fern'andez-Montequ'ın 2009 | Cuba | 101/48 | 76/73 | EGF vs SOC | 65.2 | 64.0 | 1 year | 8 weeks | Wagner: 3–4 | ①④ |
| Gomez-Villa 2014 | Mexico | 17/17 | 21/13 | EGF vs SOC | 62.1 | 55.1 | 8 weeks | 8 weeks | TEXAS: A/B | ① |
| Oliveira 2021 | Brazil | 14/11 | 18/7 | EGF vs SOC | 60.6 | 65.1 | 12 weeks | 12 weeks | — | ① |
| Park 2018 | South Korea | 82/85 | 104/63 | EGF vs SOC | 56.52 | 59.31 | 12 weeks | 12 weeks | Wagner: 1–2 | ① |

**Table 1** (*continued*)

| Author year | Country | Size I/C | Gender distribution M/F | Comparisons | Mean age (years) | | Study duration | Treatment duration | Ulcer grade | Outcomes |
|---|---|---|---|---|---|---|---|---|---|---|
| Singla 2014 | India | 25/25 | 44/6 | EGF *vs* SOC | 58.80 | 55.84 | 8 weeks | 8 weeks | Wagner: 1–2 | ① |
| Tsang 2003 | China | 42/19 | 29/32 | EGF *vs* SOC | 65.5 | 64.37 | 24 weeks | 12 weeks | Wagner: 1–2 | ①④ |
| Viswanathan 2020 | India | 27/23 | 27/23 | EGF *vs* SOC | 57.9 | 55.0 | 30 days | 30 days | Wagner: 1–2 | ①② |
| Xu 2018 | China | 50/49 | 50/49 | EGF *vs* SOC | 65 | 63 | 8 weeks | 8 weeks | Wagner: 2 | ② |
| Zhang 2021 | China | 40/40 | 38/42 | EGF *vs* SOC | 59 | 60 | 4 weeks | 4 weeks | Wagner: 2–3 | ② |
| Jeppesen 2016 | Denmark | 11/12 | 16/7 | ESWT *vs* SOC | 65.3 | 67.8 | 7 weeks | 3 weeks | Wagner: 1–2 | ③ |
| Moretti 2009 | Italy | 15/15 | 16/14 | ESWT *vs* SOC | 56.8 | | 20 weeks | 6 days | — | ①② |
| Nossair 2013 | Egypt | 20/20 | 29/11 | ESWT *vs* SOC | 56.6 | 55.15 | 12 weeks | 3 weeks | Wagner: 2–3 | ③ |
| Omar 2014 | Saudi Arabia | 19/19 | 27/11 | ESWT *vs* SOC | 56.59 | 57.0 | 20 weeks | 8weeks | TEXAS: 1A/2A | ①②③ |
| Snyder 2018 | USA | 172/164 | 269/67 | ESWT *vs* SOC | 59.9 | 56.4 | 24 weeks | 12 weeks | TEXAS: 1A/2A | ① |
| Vangaveti 2023 | Australia | 25/23 | 34/14 | ESWT *vs* SOC | 62 | 62 | 6 weeks | 6 weeks | TEXAS: 1A or higher | ① |
| Wang 2009 | China | 34/36 | — | ESWT *vs* HBOT | 58.6 | 63.4 | 6 weeks | 6 weeks | — | ① |
| Wang 2011 | China | 39/38 | — | ESWT *vs* HBOT | 60.51 | 62.45 | 6 weeks | 4 weeks | Wagner: 2–4 | ① |
| Ennis 2005 | USA | 27/28 | — | LFU *vs* SOC | — | | 12 weeks | 12 weeks | Wagner: 1–2 | ①② |
| Abd El Fattah 2018 | Egypt | 23/23 | 24/22 | LFU *vs* SOC | 32- 58 | 35- 67 | 12 weeks | 12 weeks | TEXAS: 1–2A, 1–3B | ①③ |
| Lázaro Martínez 2020 | Spain | 27/24 | 48/3 | LFU *vs* SOC | 64.1 | 58 | 24 weeks | 6 weeks | TEXAS: 1–2B/D | ①② |
| Michailidis 2018 | Australia | 10 | — | LFU *vs* SOC | — | | 24 weeks | 24 weeks | TEXAS: 1–2A/B/C | ①② |
| Rastogi 2019 | India | 34/26 | — | LFU *vs* SOC | 52.5 | 51.2 | 4 weeks | 4 weeks | Wagner: 2–3 | ①③ |
| Baker 1997 | USA | 41/39 | 55/25 | ES *vs*SOC | 54.1 | 51.5 | 4 weeks | 4 weeks | — | ① |
| Lundeberg 1992 | Sweden | 32/32 | 26/36 | ES *vs* SOC | 67.5 | 66 | 12 weeks | 12 weeks | — | ① |
| Peters 2001 | USA | 20/20 | — | ES *vs* SOC | 54.4 | 59.9 | 12 weeks | 12 weeks | TEXAS: 1A/2A | ① |
| Malekpour Alamdari 2021 | Iran | 30/30 | 33/27 | NPWT *vs* SOC | 70.31 | 71.80 | 12 weeks | — | Wagner: 2 | ④ |
| Anjum 2022 | Pakistan | 20/20 | 28/12 | NPWT *vs* SOC | 42.95 | 46.30 | 10 weeks | 8 weeks | Wagner: 1–2 | ①② |
| Blume 2008 | USA | 169/166 | 263/72 | NPWT *vs* SOC | 58 | 59 | 36 weeks | 16 weeks | Wagner: 2–3 | ①④ |
| Karatepe 2011 | Turkey | 30/37 | 19/48 | NPWT *vs* SOC | 68.5 | 66.3 | — | — | — | ② |
| Maranna 2021 | India | 22/23 | 33/12 | NPWT *vs* SOC | 50.23 | 49.00 | 12 weeks | 2 weeks | Wagner: 1–2 | ①② |
| Nain 2011 | India | 15/15 | 25/5 | NPWT *vs* SOC | 61.33 | 55.40 | 8 weeks | 8 weeks | — | ① |
| Seidel 2020 | Germany | 171/174 | 267/78 | NPWT *vs* SOC | 67.6 | 68.1 | 24 weeks | 16 weeks | Wagner: 1-3 | ①④ |
| Seidel 2022 | Germany | 44/110 | 113/41 | NPWT *vs* SOC | 66.5 | 67.8 | 24 weeks | 16 weeks | Wagner: 2–4 | ①②④ |
| Ahmed 2017 | Egypt | 28/28 | 38/18 | PRP *vs* SOC | 43.2 | 49.8 | 12 weeks | 12 weeks | TEXAS: 1–2A/C | ① |
| Malekpour Alamdari 2021 | Iran | 43/47 | 56/34 | PRP *vs* SOC | 56.3 | 56.7 | 24 weeks | 3 weeks | Wagner: 1–2 | ② |
| Driver 2006 | USA | 40/32 | 59/13 | PRP *vs* SOC | — | | 24 weeks | 12 weeks | TEXAS: 1A | ①② |
| Elsaid 2020 | Egypt | 12/12 | 14/10 | PRP *vs* SOC | 54.7 | 55.6 | 20 weeks | 20 weeks | — | ①② |
| Gowsick 2023 | India | 87/87 | 104/70 | PRP *vs* SOC | — | | 12 weeks | 12 weeks | — | ①③ |
| Gupta 2021 | India | 30/30 | 41/19 | PRP *vs* SOC | 56.03 | 55.76 | 6 weeks | 6 weeks | Wagner: 1–2 | ①③ |
| Hossam 2022 | Egypt | 40/40 | 62/18 | PRP *vs* SOC | 54.9 | 54.8 | 12 weeks | 12 weeks | Wagner: 1–2 | ①④ |
| Li 2015 | China | 59/58 | 75/42 | PRP *vs* SOC | 61.4 | 64.1 | 12 weeks | 12 weeks | — | ① |
| Orban 2022 | Egypt | 36/36 | 41/31 | PRP *vs* SOC | 56.03 | 58.69 | 20 weeks | 20 weeks | — | ①② |
| Singh 2018 | Etawah | 29/26 | 34/21 | PRP *vs* SOC | 53.76 | 55.69 | 4 weeks | 4 weeks | — | ①②④ |
| Xie 2020 | China | 25/23 | 27/21 | PRP *vs* SOC | 60.50 | 61.10 | 8 weeks | 8 weeks | — | ①② |

**Table 1** (*continued*)

| Author year | Country | Size I/C | Gender distribution M/F | Comparisons | Mean age (years) | | Study duration | Treatment duration | Ulcer grade | Outcomes |
|---|---|---|---|---|---|---|---|---|---|---|
| Chen 2017 | China | 20/18 | 21/17 | HBOT *vs* SOC | 64.3 | 60.8 | 6 weeks | 4 weeks | Wagner: 1-3 | ①④ |
| Faglia 1996 | Italy | 35/33 | 48/20 | HBOT *vs* SOC | 61.7 | 65.6 | — | Mean session = 38.8 ± 8 | Wagner: 2–4 | ④ |
| Abidia 2003 | UK | 9/9 | 9/9 | HBOT *vs* SOC | 72 | 70 | 1 year | 6 weeks | Wagner: 1–2 | ①④ |
| Duzgun 2008 | Turkey | 50/50 | 64/36 | HBOT *vs* SOC | 58.1 | 63.3 | 92 ± 12 weeks | 6 weeks | Wagner: 2–4 | ①④ |
| Fedorko 2016 | Canada | 49/54 | 69/34 | HBOT *vs* SOC | 61 | 62 | 12 weeks | 6 weeks | Wagner: 2–4 | ①③④ |
| Kessler 2003 | France | 14/13 | 19/8 | HBOT *vs* SOC | 60.2 | 67.6 | 6 weeks | 2 weeks | Wagner: 1-3 | ①③ |
| Kumar 2020 | India | 28/26 | 39/15 | HBOT *vs* SOC | 58.4 | 56.9 | 1 year | 6 weeks | Wagner: 2–4 | ①④ |
| Londahl 2010 | Sweden | 49/45 | 56/38 | HBOT *vs* SOC | 69 | 68 | 1 year | 8 weeks | Wagner: 2–4 | ①④ |
| Salama 2019 | Egypt | 15/15 | 22/8 | HBOT *vs* SOC | 55.1 | 57.7 | 16 weeks | 4-8 weeks | Wagner: 2–3 | ①④ |
| Santema 2018 | Netherlands | 60/60 | 97/23 | HBOT *vs* SOC | 67.6 | 70.6 | 1 year | 8 weeks | Wagner: 2–4 | ①④ |
| Al-Jalodi 2022 | USA | 81/64 | — | TOT *vs* SOC | — | | 1 year | 12 weeks | ISDA: 1 -2 | ①④ |
| Driver 2017 | USA | 65/63 | 95/33 | TOT *vs* SOC | 59 | | 12 weeks | 12 weeks | TEXAS: 1A | ① |
| Frykberg 2020 | UK | 36/37 | 63/10 | TOT *vs* SOC | 64.6 | 61.9 | 1 year | 12 weeks | TEXAS: 1–2 | ①②④ |
| He 2021 | China | 40/40 | 49/31 | TOT *vs* SOC | 62.7 | 63.1 | 1 year | 8 weeks | Wagner: 2–4 | ①②③④ |
| Niederauer 2015 | USA | 21/21 | 33/9 | TOT *vs* SOC | 58.3 | 59.2 | 12 weeks | 12 weeks | — | ① |
| Niederauer 2017 | USA | 50/50 | 79/21 | TOT *vs* SOC | 57.5 | 59.1 | 12 weeks | 12 weeks | — | ① |
| Niederauer 2018 | USA | 74/72 | 113/33 | TOT *vs* SOC | 56.1 | 56.6 | 12 weeks | 12 weeks | TEXAS: 1A | ① |
| Serena 2021 | UK | 81/64 | 107/37 | TOT *vs* SOC | 64.20 | 62.69 | 12 weeks | 12 weeks | Wagner: 1–2 | ①③ |
| Yu 2016 | Canada | 10/10 | 17/3 | TOT *vs* SOC | 57 | 58 | 8 weeks | 8 weeks | — | ① |
| Didomenico 2018 | USA | 40/40 | 54/26 | AMT *vs* SOC | 60.1 | 61.0 | 12 weeks | 12 weeks | — | ①② |
| Game 2021 | UK | 15/16 | 25/6 | AMT *vs* SOC | 62.8 | 57 | 12 weeks | 12 weeks | — | ①④ |
| Lavery 2014 | USA | 50/47 | 68/29 | AMT *vs* SOC | 55.5 | 55.1 | 12 weeks | 12 weeks | — | ①④ |
| Mohajeri-Tehrani 2016 | Iran | 27/30 | 37/20 | AMT *vs* SOC | 55.44 | 60 | 6 weeks | 6 weeks | Wagner: 2–4 | ①④ |
| Serena 2019 | USA | 38/38 | 59/17 | AMT *vs* SOC | 59.2 | 59.6 | 16 weeks | 12 weeks | Wagner: 1–2 | ① |
| Snyder 2016 | USA | 15/14 | 25/4 | AMT *vs* SOC | 57.9 | 58.6 | 6 weeks | 6 weeks | Wagner: 1–2 | ① |
| Tettelbach 2019a | USA | 54/56 | 80/30 | AMT *vs* SOC | 57.4 | 57.1 | 16 weeks | 12 weeks | — | ① |
| Tettelbach 2019b | USA | 101/54 | 126/29 | AMT *vs* SOC | 58.3 | 56.3 | 16 weeks | 12 weeks | — | ① |
| Thompson 2019 | USA | 7/6 | 11/2 | AMT *vs* SOC | 58.50 | 55.17 | 16 weeks | 12 weeks | TEXAS: 1 | ① |
| Zelen 2013 | USA | 13/12 | 16/9 | AMT *vs* SOC | 56.4 | 61.7 | 12 weeks | 6 weeks | — | ①② |
| Zelen 2015 | USA | 20/20 | 19/21 | AMT *vs* SOC | 63.2 | 62.2 | 12 weeks | 6 weeks | — | ① |
| Zelen 2016 | USA | 32/35 | 41/26 | AMT *vs* SOC | 63.3 | 60.6 | 12 weeks | 6 weeks | — | ①② |

**Notes.**

① Wound healing rate; ② Wound healing time; ③ percentage area reduction (PAR); ④ Amputation.

*Afshari et al., 2005; Fernández-Montequín et al., 2009; Gomez-Villa et al., 2014; Oliveira et al., 2021; Park et al., 2018; Singla et al., 2014; Tsang et al., 2003; Viswanathan, Juttada & Babu, 2020; Xu et al., 2018; Zhang et al., 2021a; Agrawal et al., 2009; Bhansali et al., 2009; Blume et al., 2011; Jaiswal et al., 2010; Khandelwal et al., 2013; Ma et al., 2015; Samuel et al., 2016; Steed, 1995; Wieman, Smiell & Su, 1998; Ahmed et al., 2017; Driver et al., 2006; Elsaid et al., 2020; Gowsick et al., 2023; Gupta et al., 2021; Hossam et al., 2022; Li et al., 2015; Malekpour Alamdari et al., 2021b; Orban et al., 2022; Singh et al., 2018; Xie et al., 2020; Lu et al., 2008; Han, Kim & Kim, 2010; Huang et al., 2005; Jain et al., 2011; Lu et al., 2011; Mohammadzadeh et al., 2013; Ozturk et al., 2012; Uzun et al., 2021; Abd El Fattah et al., 2018; Ennis et al., 2005; Lázaro-Martínez et al., 2020; Michailidis et al., 2018; Rastogi, Bhansali & Ramachandran, 2019; Anjum et al., 2022; Blume et al., 2008; Karatepe et al., 2011; Malekpour Alamdari et al., 2021a; Maranna et al., 2021; Nain et al., 2011; Seidel, Lefering & DiaFu study, 2022; Seidel et al., 2020; Haze et al., 2022; Kaviani et al., 2011; Minatel et al., 2009; De Alencar Fonseca Santos et al., 2018; Naidu et al., 2005; Baker et al., 1997; Lundeberg, Eriksson & Malm, 1992; Peters et al., 2001; Jeppesen et al., 2016; Moretti et al., 2009; Nossair, Eid & Salama, 2013; Omar et al., 2014; Sandoval Ortíz et al., 2014; Snyder et al., 2018; Vangaveti et al., 2023; Wang et al., 2009; Wang, Wu & Yang, 2011; DiDomenico et al., 2018; Game et al., 2021; Lavery et al., 2014; Serena et al., 2020; Snyder et al., 2016; Tettelbach et al., 2019a; Tettelbach et al., 2019b; Thompson et al., 2019; Zelen et al., 2015; Zelen et al., 2013; Zelen et al., 2016; Abidia et al., 2003; Chen et al., 2017; Duzgun et al., 2008; Faglia et al., 1996; Fedorko et al., 2016; Kessler et al., 2003; Kumar et al., 2020; Londahl et al., 2010; Salama et al., 2019; Santema et al., 2018; Al-Jalodi et al., 2022; Driver et al., 2017; Frykberg et al., 2020; He et al., 2021; Niederauer, Michalek & Armstrong, 2015; Niederauer, Michalek & Armstrong, 2017; Niederauer et al., 2018; Serena et al., 2021; Yu et al., 2016.*

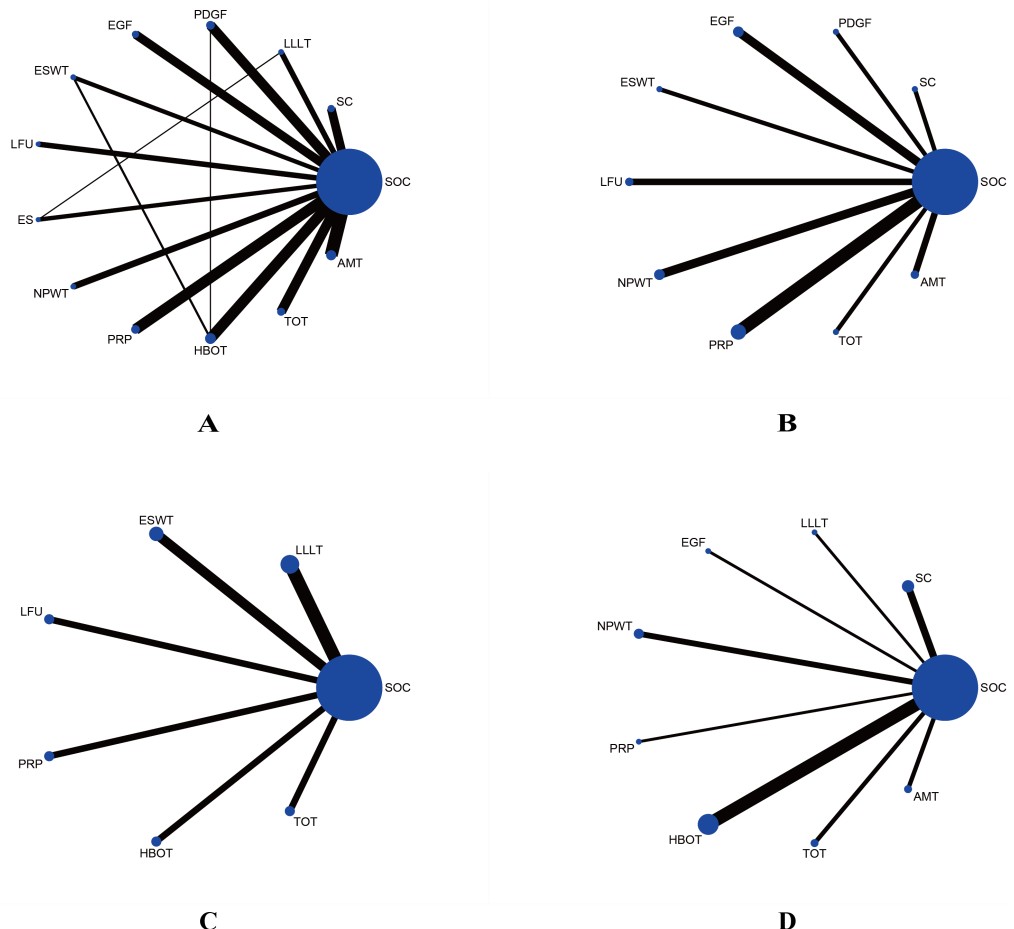

**Figure 2  Network plots comparing different interventions for four outcomes.** (A) Wound healing rate; (B) wound healing time; (C) percentage area reduction (PAR); (D) amputation rate. The blue nodes represent individual interventions, with node size proportional to the sample size of each intervention. Lines between nodes indicate direct comparisons from included RCTs, with line thickness proportional to the number of studies performing each comparison.

## Quality of the included studies

According to the Cochrane Collaboration standards, almost half of the studies failed to report the blinding of researchers, participants, and medical assessors, and the allocation concealment was not disclosed. 17 studies reported a high risk of bias because they were not blinded (Fig. 3, Supplemental Information 4).

## Wound healing rate

This network meta-analysis evaluated 12 interventions for the treatment of DFU, emphasizing wound healing rate. Based on 95 RCTs involving 6,807 patients, our analysis revealed that, compared with SOC, only LFU (OR = 2.20; 95% CI [0.99–4.91]) and ES (OR = 1.88; 95% CI [0.87–4.05]) among the 12 evaluated interventions failed to achieve a statistically significant enhancement in wound healing rate. The remaining 10 interventions, including SC, AMT, LLLT, EGF, PDGF, ESWT, NPWT, PRP, HBOT, and

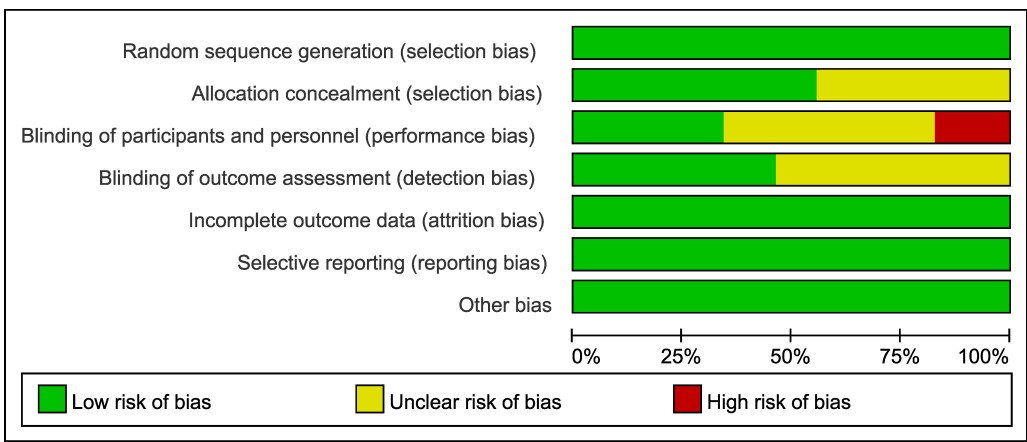

**Figure 3  Risk of bias graph of the included studies.** The proportion of studies rated as having low, unclear, or high risk of bias across seven domains is presented.

TOT, significantly enhanced wound healing rate. In pairwise comparisons among the 12 interventions, SC significantly improved wound healing rate compared with PDGF (OR = 0.33; 95% CI [0.13–0.83]), ES (OR = 0.33; 95% CI [0.11–0.97]), and HBOT (OR = 0.39; 95% CI [0.16–0.97]). Additionally, AMT significantly outperformed PDGF (OR = 2.70; 95% CI [1.34–5.45]), ES (OR = 2.72; 95% CI [1.10–6.68]), HBOT (OR = 2.31; 95% CI [1.15–4.62]), and TOT (OR = 2.13; 95% CI [1.11–4.10]) in improving healing outcomes (see Fig. 4). According to the SUCRA rankings, SC (89.7%) and AMT (89.2%) ranked highest, while PDGF (25.7%) and SOC (0.8%) ranked lowest (see Fig. 5A). Supplemental Information 3 displays the ratios of different interventions for wound healing rate, with statistically significant findings highlighted in blue.

## Wound healing time

This network meta-analysis examined nine interventions for DFU: AMT, PRP, ESWT, NPWT, SC, PDGF, TOT, LFU, and EGF, primarily focusing on their mean healing times. Based on 28 RCTs involving 1,591 patients, our analysis demonstrated that, compared with SOC, AMT (MD = −26.91 days; 95% CI [−44.27 to −9.55]), PRP (MD = −21.65 days; 95% CI [−33.61 to −9.69]), and NPWT (MD = −16.79 days; 95% CI [−31.12 to −2.26]) significantly reduced woundr healing time. Among 10 interventions, AMT (SUCRA = 84.7%) and PRP (SUCRA = 74.6%) ranked highest according to SUCRA values, while LFU (29.4%) and SOC (10.4%) ranked lowest (see Figs. 5B, 6).

## PAR

This network meta-analysis evaluated six interventions for DFU: LLLT, ESWT, PRP, HBOT, LFU, and TOT, primarily focusing on PAR. Based on 15 RCTs involving 880 patients, our analysis revealed that, compared with SOC, LLLT (MD = 34.27; 95% CI: 17.35 to 51.20) and ESWT (MD = 27.50; 95% CI [11.00–44.00]) significantly reduced the PAR. Among the seven interventions evaluated, LLLT demonstrated a significantly greater reduction in ulcer area compared with both HBOT (MD = −31.42, 95% CI [−58.74

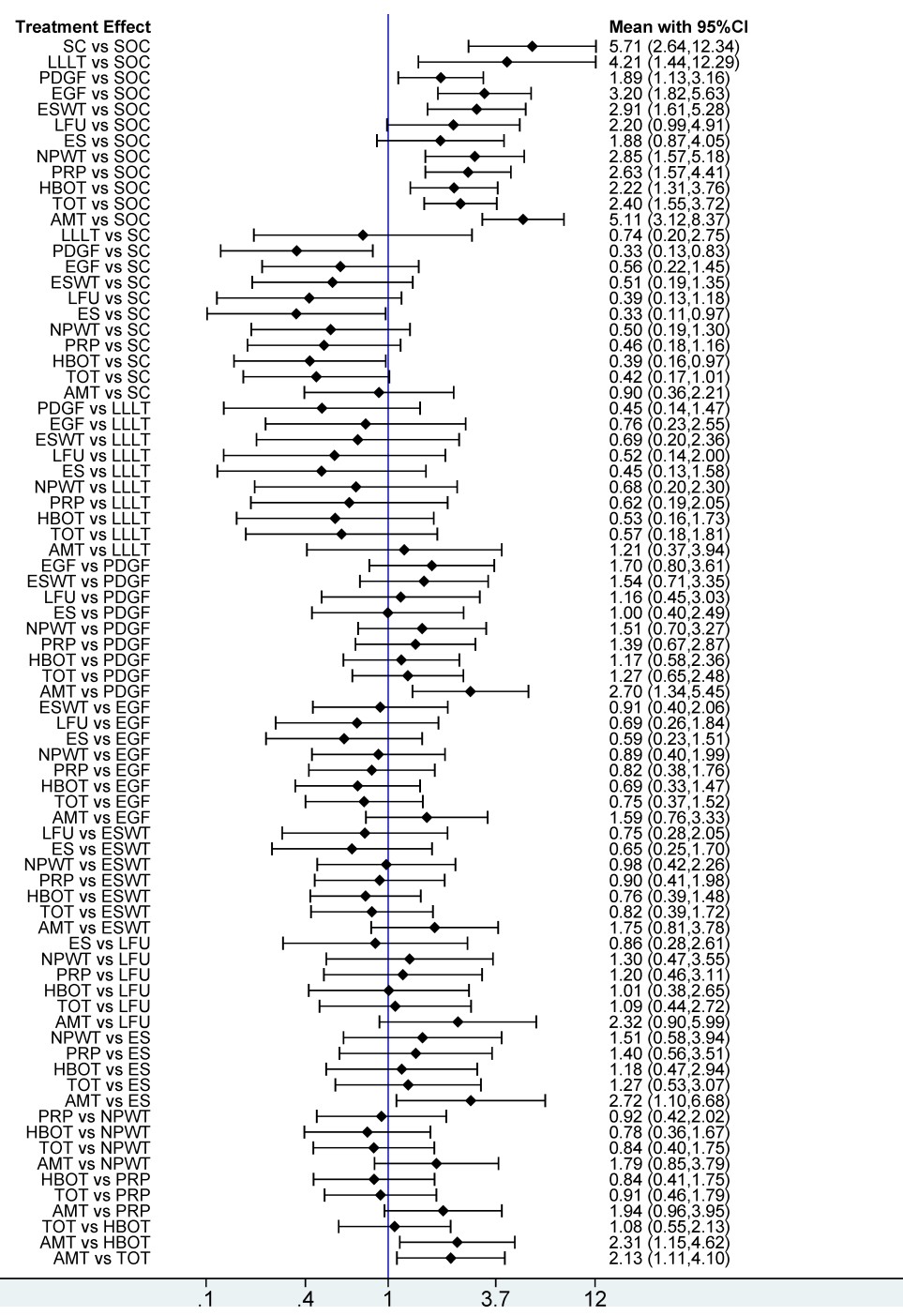

**Figure 4** **Forest plot of network meta-analysis results for wound healing rate.** Treatment effects are presented as odds ratios (ORs) with 95% confidence intervals (CIs).

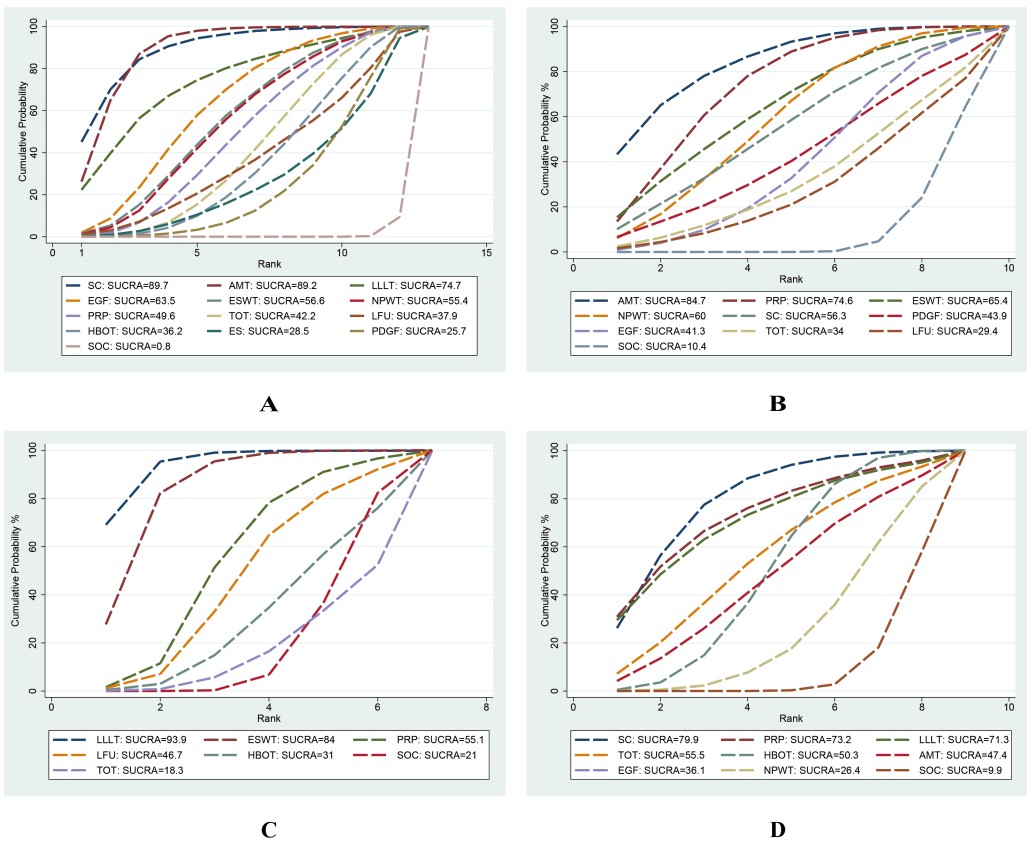

**Figure 5 Surface Under the Cumulative Ranking (SUCRA) values of different interventions for four outcomes.** (A) Wound healing rate; (B) wound healing time; (C) percentage area reduction (PAR); (D) amputation rate. Higher SUCRA values and larger areas under the curve indicate a greater probability that the intervention is more effective for the corresponding outcome.

to −4.10]) and TOT (MD = −36.93, 95% CI [−63.54 to −10.31]). In addition, ESWT also significantly reduced PAR compared with TOT (MD = −30.15; 95% CI [−56.27 to −4.03]). According to the SUCRA rankings, LLLT (93.9%) and ESWT (84.0%) ranked highest, while SOC (21.0%) and TOT (18.3%) ranked lowest (see Figs. 5C, 7A).

## Amputation rate

This network meta-analysis evaluated eight interventions for DFU: PRP, LLLT, SC, TOT, HBOT, AMT, EGF, and NPWT, primarily focusing on amputation rate. Based on 30 RCTs involving 2,526 patients, our analysis revealed that, compared with SOC, SC (OR = 0.12; 95% CI [0.03–0.55]) and HBOT (OR = 0.35; 95% CI [0.16–0.78]) significantly reduced the amputation rate. According to the SUCRA rankings, SC (79.9%) and PRP (73.2%) ranked highest, whereas NPWT (26.4%) and SOC (9.9%) ranked lowest (refer to Figs. 5D, 7B).

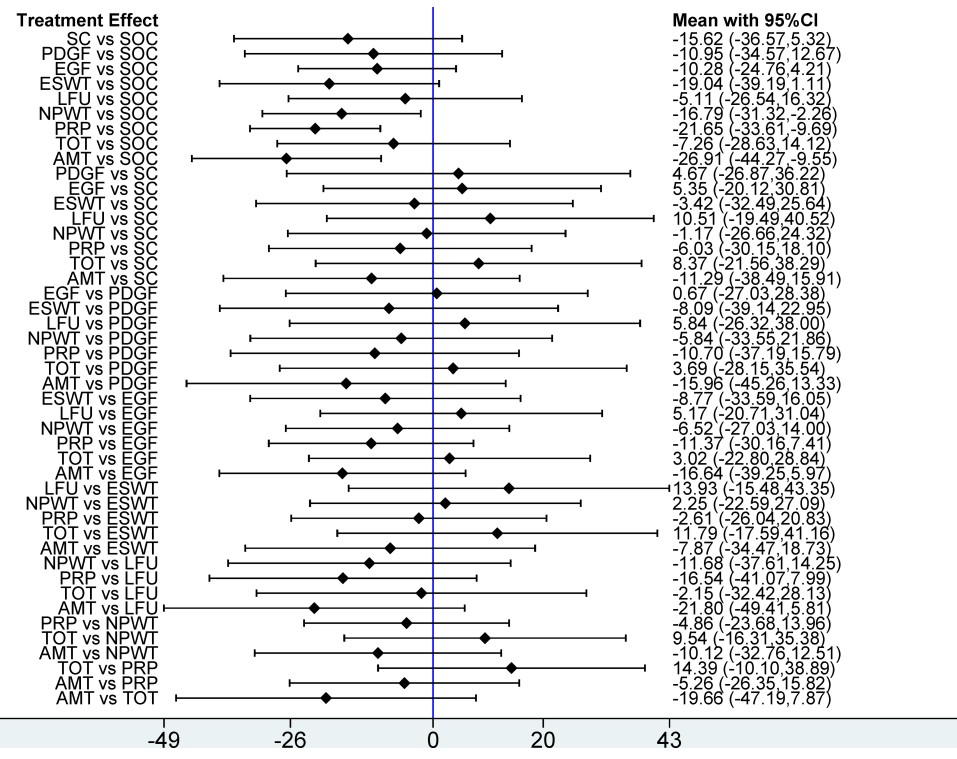

**Figure 6** **Forest plot of network meta-analysis results for wound healing time.** Treatment effects are presented as mean differences (MDs) with 95% confidence intervals (CIs).

## Subgroup analysis

We performed a subgroup analysis based on follow-up duration: subgroup 1 included studies with follow-up ≤12 weeks, and subgroup 2 included those with follow-up >12 weeks. Regarding wound healing rate, neither ESWT nor HBOT showed a significant benefit compared to SOC within 12 weeks. However, beyond 12 weeks, both ESWT (OR = 2.67; 95% CI [1.06–6.74]) and HBOT (OR = 4.99; 95% CI [1.73–14.36]) significantly improved healing rate. For amputation rate, HBOT did not show a significant reduction within 12 weeks compared to SOC, but after more than 12 weeks, it significantly reduced the amputation rate (OR = 0.28; 95% CI [0.10–0.79]) (see Fig. 8).

## Inconsistency assessment

We evaluated inconsistency using global inconsistency tests and a node-splitting approach for all comparisons. The inconsistency tests indicated no significant inconsistency in this NMA ($p = 0.2268$). Outcomes from the node-splitting analysis, presented in Supplemental Information 3, indicate no inconsistency between direct and indirect comparisons.

## Small-scale study effect analysis

Our analysis revealed no indication that smaller-scale studies influenced various outcomes (Fig. 9).

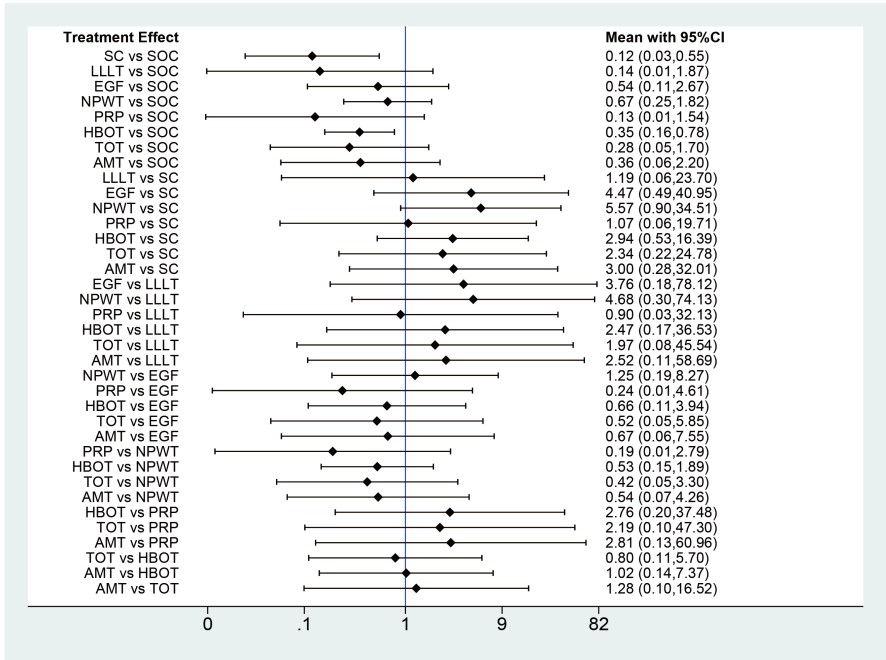

**A**

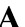

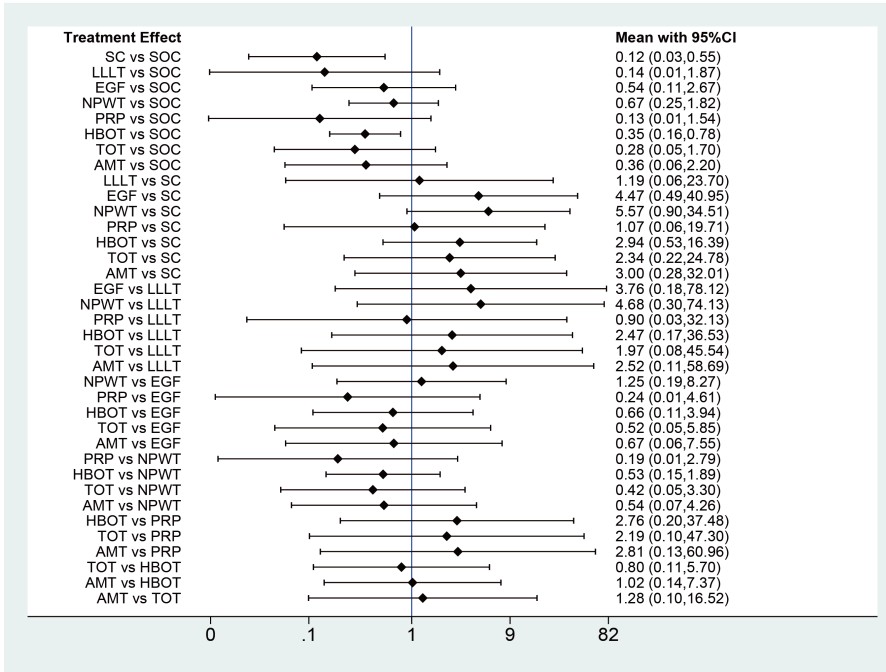

**B**

**Figure 7** **Forest plots of network meta-analysis results for (A) percentage area reduction (PAR) and (B) amputation rate.** Treatment effects are presented as mean differences (MDs) for PAR and odds ratios (ORs) for amputation rate, with 95% confidence intervals (CIs).

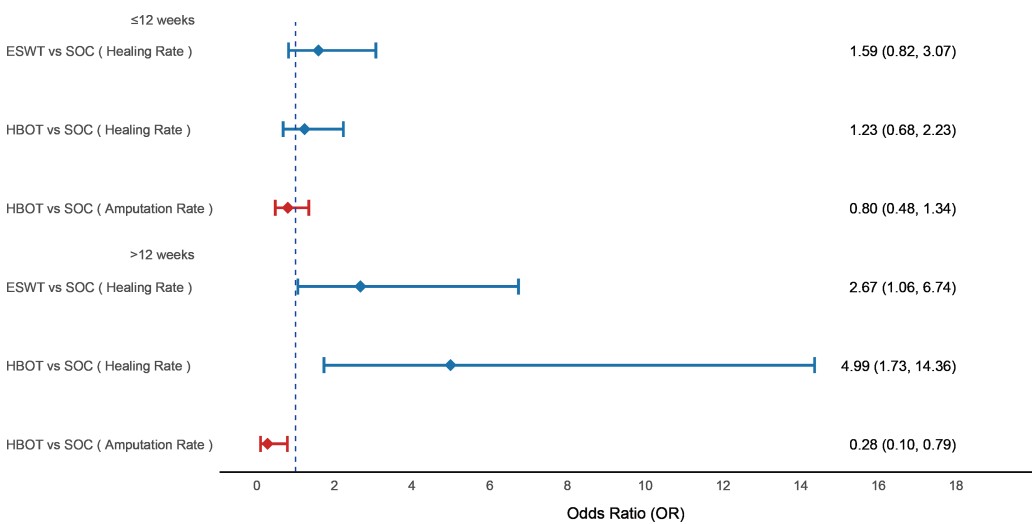

**Figure 8  Forest plot of subgroup analysis based on follow-up duration.** Subgroup 1 included studies with follow-up ≤ 12 weeks, and subgroup 2 included studies with follow-up >12 weeks. Treatment effects are presented as odds ratios (ORs) with 95% confidence intervals (CIs).

## DISCUSSION

This study incorporated 99 RCTs involving 7,356 patients and utilized NMA to integrate direct and indirect comparisons (*Rouse, Chaimani & Li, 2017*), enhancing statistical power to systematically evaluate the effectiveness of 12 DFU treatments. Primary outcome measures included wound healing rate, wound healing time, PAR, and amputation rate. All interventions were compared against SOC or placebo. Our analysis showed that, compared to SOC, ten interventions—including SC, AMT, LLLT, EGF, PDGF, ESWT, NPWT, PRP, HBOT, and TOT—significantly improved wound healing rate. SC significantly outperformed PDGF, ES, and HBOT in promoting wound healing. Moreover, AMT was significantly more effective than PDGF, ES, HBOT, and TOT, with SC ranked highest according to SUCRA. In terms of wound healing time, AMT, PRP, and NPWT significantly reduced healing duration compared to SOC, with LLLT ranking highest in SUCRA. Regarding ulcer area reduction, both LLLT and ESWT were significantly superior to SOC. Furthermore, LLLT significantly outperformed HBOT and TOT, and ESWT also showed superiority over TOT, with LLLT achieving the highest SUCRA ranking. For amputation rate, SC and HBOT significantly reduced the risk compared to SOC, with SC ranked highest according to SUCRA.

In this NMA, we observed that multiple interventions demonstrated potential advantages in promoting DFU healing, particularly SC, AMT, and LLLT. Among these, the effect of SC was generally consistent with previous studies: multiple meta-analyses have confirmed that stem cells not only significantly improve wound healing rate and reduce the risk of amputation but also positively impact patients' quality of life (*Dai et al., 2020*; *Elsharkawi et al., 2023*; *Guo et al., 2017*; *Mudgal et al., 2024*; *Shu et al., 2018*; *Sun et al., 2022c*; *Zhang, Deng & Tang, 2017*). Compared with SOC, SC demonstrated superior clinical performance

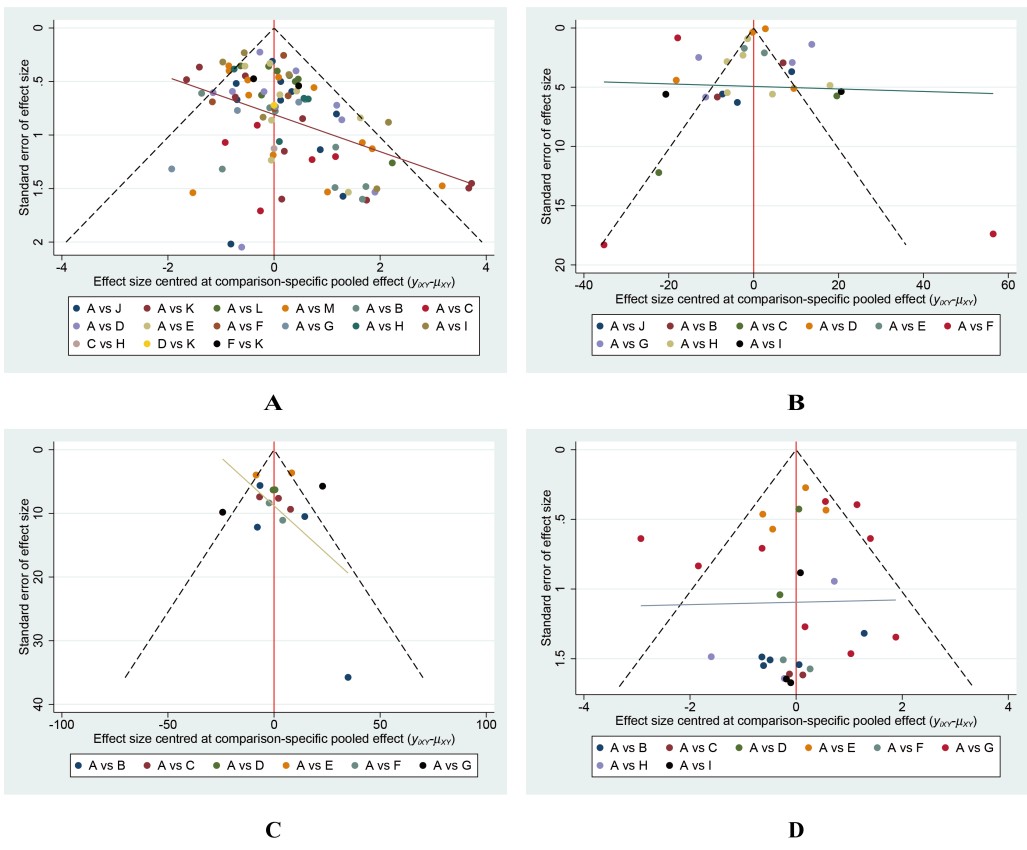

**Figure 9  Funnel plots of the study treatments.** (A) Wound healing rate; (B) Wound healing time; (C) Percentage area reduction (PAR); (D) Amputation rate.

in this study. Notably, our study is the first to reveal that SC was more effective than PDGF and HBOT in improving wound healing rate, which may be attributed to the multiple biological mechanisms of stem cells. In contrast, PDGF and HBOT exhibit certain limitations. PDGF promotes fibroblast proliferation and collagen synthesis, but its short half-life and rapid degradation limit sustained activity, and its isolated signaling lacks the broad paracrine effects of stem cells (*Sun et al., 2024*; *Zhang et al., 2021b*). HBOT enhances angiogenesis by increasing oxygen availability, but its indirect effects depend on patient-specific factors, and it lacks the multilineage differentiation and immunomodulatory capabilities of stem cells (*Goldstein, 2013*; *Peña-Villalobos et al., 2018*). The superior performance of SC may be explained by its comprehensive mechanisms, including the release of angiogenic growth factors, modulation of inflammation and immune responses, reduction of fibrosis, and optimization of the wound microenvironment through extracellular matrix remodeling (*Panda & Nayak, 2024*). Owing to their pluripotency and diverse paracrine and chemotactic effects, stem cells exhibit robust tissue regeneration potential both locally and systemically. Two recent NMAs involving SC compared its efficacy with that of PRP and EGF and found no significant differences (*Yang et al., 2025*). Another NMA included 11 different interventions and highlighted the superiority of SC

over SOC, but did not find SC to be superior to other interventions or effective in reducing the risk of amputation (*OuYang et al., 2024*). These differences may be attributed to the scope of included studies, patient characteristics, follow-up duration, and stem cell types. The stem cell therapies included in this study primarily involve bone marrow-derived stem cells, adipose-derived stem cells, and peripheral blood-derived stem cells. Stem cells commonly used to treat DFU in clinical practice are categorized into two types: somatic stem cells and pluripotent stem cells. Among these, mesenchymal stem cells (MSCs) are the most commonly used form of somatic stem cells, with bone marrow-derived MSCs being the preferred cell type for clinical applications in diabetic wounds. Compared with bone marrow-derived MSCs, adipose-derived MSCs exhibit similar potential to enhance wound healing, and clinical research on peripheral blood stem cells is also increasing (*Ho et al., 2023*).

Second, consistent with previous studies, we found that AMT was significantly superior to SOC in improving wound healing rate and shortening wound healing time. Several meta-analyses demonstrated its effectiveness in significantly improving DFU healing rate and markedly reducing healing time (*Haugh et al., 2017*; *Huang et al., 2020a*; *Laurent et al., 2017*; *Mohammed et al., 2022*; *Su et al., 2020*). We also found that AMT was significantly superior to PDGF, ES, HBOT, and TOT in improving wound healing rate, potentially representing the first time this conclusion has been reported. A previous NMA of nine different dressings also demonstrated that amniotic membrane dressings had the greatest advantage in healing rate among multiple interventions (*Zhang, Sun & Jiang, 2019*). AMT's therapeutic mechanisms involve abundant growth factors and key cellular components present in amniotic membranes, offering anti-inflammatory, anti-fibrotic, and angiogenesis-promoting effects. As a natural biological scaffold, AMT facilitates tissue regeneration and cell proliferation, especially beneficial in clinical scenarios requiring rapid wound closure (*Koob et al., 2013*; *Munoz-Torres et al., 2022*; *Xiao et al., 2021*). Third, we observed that LLLT was significantly superior to SOC in improving wound healing rate and reducing wound area, consistent with the findings of existing meta-analyses (*Huang et al., 2021*; *Li et al., 2018*; *Santos et al., 2021*; *Zhou et al., 2021*). Notably, this study found that LLLT was also superior to HBOT and TOT in reducing ulcer area. Currently, no NMA has systematically compared the efficacy of LLLT with multiple other interventions. LLLT promotes wound healing through photobiomodulation, significantly enhancing cellular metabolism, improving microcirculation, and stimulating collagen synthesis to expedite wound repair (*Frangez et al., 2017*; *Peplow & Baxter, 2012*; *Tatmatsu-Rocha et al., 2016*).

Subgroup analysis revealed that HBOT had no significant effect on healing rate and amputation rate within 12 weeks but demonstrated significant therapeutic effects after 12 weeks. This finding is consistent with previous observations suggesting the long-lasting effectiveness of HBOT in certain pathophysiological conditions involving irradiated tissues (*Marx et al., 1990*). One RCT also failed to detect significant differences at 12 weeks between HBOT and SOC (*Fedorko et al., 2016*). HBOT enhances tissue oxygenation, promotes collagen synthesis and angiogenesis, significantly reducing amputation risk (*Huang et al., 2020b*; *Zhang et al., 2022*). Similarly, ESWT did not demonstrate a healing advantage in the short term but significantly improved healing rate in follow-up periods

longer than 12 weeks. One RCT found no significant difference in wound healing between the ESWT group and SOC during the first 12 weeks; however, a statistically significant difference emerged at week 20 (*Snyder et al., 2018*). ESWT accelerates tissue regeneration and local angiogenesis *via* mechanical stimulation (*Cao et al., 2025*). These findings suggest that the therapeutic effects of HBOT and ESWT are time-dependent, with cumulative benefits emerging after prolonged treatment, particularly beyond 12 weeks. However, no meta-analysis has yet conducted subgroup analyses of HBOT and ESWT based on follow-up duration, and this research gap warrants further attention.

Clarifying the specific advantages and clinical contexts of different therapies may assist clinicians in making personalized and staged treatment decisions. For patients who do not respond to SOC, priority should be given to SC, AMT, or LLLT. In particular, SC and AMT are more suitable for severe or refractory DFUs, such as neuroischemic ulcers, and can rapidly improve the wound surface in the early stages, whereas LLLT is better suited for outpatient clinics or resource-limited settings due to its high safety profile, relatively low cost, and ability to accelerate wound healing. Additionally, HBOT and ESWT demonstrate greater effectiveness in follow-up periods exceeding 12 weeks. HBOT can provide effective microcirculatory improvement for chronic ischemic ulcers, whereas ESWT is beneficial for patients with more severe fibrosis. However, advanced technologies such as SC, AMT, and HBOT require substantial investment in cost, equipment, and patient compliance, and their clinical application must be tailored to available medical resources and individual patient conditions to support comprehensive decision-making.

## STUDY LIMITATION

The main limitation of this study is the lack of detailed classification of ulcer types. Most current RCTs do not provide detailed distinctions between different types of DFU (*e.g.*, neuropathic, ischemic, or mixed) in their reports, thereby precluding subgroup comparisons based on ulcer type. Future clinical studies should improve the reporting of DFU types and conduct more detailed population stratification analyses to provide more precise evidence-based support for individualized treatment strategies.

## CONCLUSION

In this NMA, we assessed 12 interventions for DFU, focusing on outcomes such as healing rate, wound healing times, PAR, and amputation rate. SC and AMT emerged as highly effective, significantly improving healing rate compared to PDGF, ES, and HBOT. SC was also associated with reduced amputation rate, while AMT significantly shortened wound healing time. LLLT exhibited considerable effectiveness in reducing ulcer areas. The therapeutic benefits of HBOT and ESWT appeared to be time-dependent, with greater effectiveness observed after 12 weeks. These results support a more individualized approach in treating DFU, where therapy selection can be customized based on specific treatment efficacies.

### Funding

The authors received no funding for this work.

### Competing Interests

The authors declare there are no competing interests.

### Author Contributions

- Xuyang Hu conceived and designed the experiments, performed the experiments, prepared figures and/or tables, and approved the final draft.
- Huixin Meng conceived and designed the experiments, performed the experiments, analyzed the data, prepared figures and/or tables, and approved the final draft.
- Jiaru Liang conceived and designed the experiments, performed the experiments, prepared figures and/or tables, and approved the final draft.
- Hang An conceived and designed the experiments, performed the experiments, prepared figures and/or tables, and approved the final draft.
- Jiaqi Zhou analyzed the data, prepared figures and/or tables, and approved the final draft.
- Yuling Gao analyzed the data, prepared figures and/or tables, and approved the final draft.
- Chong You analyzed the data, prepared figures and/or tables, and approved the final draft.
- Zhenni Zhang analyzed the data, prepared figures and/or tables, and approved the final draft.
- Xiaoyang Gong conceived and designed the experiments, authored or reviewed drafts of the article, and approved the final draft.
- Yong Liu conceived and designed the experiments, authored or reviewed drafts of the article, and approved the final draft.

### Data Availability

The raw data is available in the Supplemental File.

### Supplemental Information

Supplemental information for this article can be found online at http://dx.doi.org/10.7717/peerj.19809#supplemental-information.

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
