# Peer review of "Comparison of the efficacy of 12 interventions in the treatment of diabetic foot ulcers: a network meta-analysis"

_PeerJ, doi:10.7717/peerj.19809_

## Round 0.1 · original submission · Major Revisions

The study entitled “Comparison of the efficacy of different interventions in the treatment of diabetic foot ulcers: a network meta-analysis” demonstrated interesting findings using an appropriate methodological approach. However, some important points must be clarified in the manuscript. Your article has great potential for publication on PeerJ, but the reviewers have requested substantial changes to be made.

Please ensure that all review, editorial, and staff comments are addressed in a response letter and that any edits or clarifications mentioned in the letter are also inserted into the revised manuscript where appropriate.

Reviewer 1 ·

Basic reporting

No comment

Experimental design

-Detailed information on the search strategy should be reported in the Prisma Checklist.
-Authors should indicate the minimum duration of the included studies.
-Authors should clarify why they chose to also include short-term studies(15-20 days).

Validity of the findings

-In discussion on line 171 there is an error: the authors report that the studies included are 118 instead of 114.
-Your discussion needs more detail.Several limitations should be acknowledged and considered in interpreting the obtained results.
The present network meta-analysis, unfortunately, included several low-quality, short-term and not adequately sized trials possibly introducing biases and affecting the overall results.
The heterogeneity of inclusion and exclusion criteria, study procedures, and definition
of endpoints could have further contributed to introduce another bias (inconsistency bias).
In particular, the main limitation of the study is having included studies conducted on different types of ulcers.
I believe that the authors should have done sub-analyses in the different types of ulcers treated in RCTS: ischemic, neuropathic and infected.

·

Basic reporting

Thank you for this opportunity to review this manuscript. I have perused all the articles and have commented that the author needs more work to improve the literature review and all the things related to the intervention of DFU. I think that the background should have more evidence to support the study. There was a lack of perspective on what they were trying to say. Also, the sentences do not flow well at times.

The topic is too broad, as I have read one of these types of review manuscripts describing intervention for DFU. if the author would like to continue this work, I think they need to write hundreds of pages of work. Of course, this is not suitable for publication standards.

Experimental design

The study is a type of literature review (meta-analysis). So, no need to provide further comments on this section.

Validity of the findings

Since the background provided the lack of literature review, I would say that this part has a concern of its validity.

Additional comments
* * *
Reviewer 3 ·

Basic reporting

No comment

Experimental design

No comment

Validity of the findings

Comments
151 3.6 Amputation rate
Does this refer to minor or major amputation or both?
Line 53
"This study employs Network Meta-Analysis (NMA) to evaluate the efficacy of fourteen interventions for DFU, enabling comprehensive treatment comparisons across multiple RCTs."
I wish it to be confirmed that the authors studied RCTS and not meta analyses as primary data

How inclusive has the selection of studies been?
Why has the following two studies not been included?
Game F, Jeffcoate W, Tarnow L, et al; LeucoPatch II Trial Team. LeucoPatch system for the management of hard-to-heal diabetic foot ulcers in the UK, Denmark, and Sweden: an observer-masked, randomised controlled trial. Lancet Diabetes Endocrinol 2018;6:870–878

.Edmonds M, Lázaro-Martínez JL, Alfayate-Garcia JM, et al Sucrose octasulfate dressing versus control dressing in patients with neuroischaemic diabetic foot ulcers (Explorer): an international, multicentre, double-blind, randomised, controlled trial. Lancet Diabetes Endocrinol 2018;6:186–196

FIGURE 2. Network of the study treatments. (A) wound healing rate; (B) wound healing
time; (C) percentage area reduction (PAR); (D) amputation rates
FIGURE 4. The surface under the cumulative ranking curve (SUCRA) for all study interventions. (A) wound healing rate; (B) wound healing time; (C) percentage area reduction (PAR); (D) amputation rates
I find these figures difficult to understand

Line 134
“Furthermore, statistical analyses revealed significant discrepancies, particularly regarding SC and AMT, compared with other treatments such as PDGF, ES, HBOT, and ozone therapy. Additionally, AMT surpassed TOT (refer to Figure 5A)”
I do not understand “discrepancy” in this context?
Line 170
“The quality of the evidence was generally characterized by a low or unclear risk of bias across 97 of the 118 studies (constituting 82%)”
I do not understand this sentence.

---

## Round 0.2 · Minor Revisions

Dear authors,

The study entitled “Comparison of the efficacy of 12 interventions in the treatment of diabetic foot ulcers: a network meta-analysis” demonstrated interesting findings using an appropriate methodological approach. However, minor revisions must be clarified in the manuscript.

Reviewer 3 ·

Basic reporting

The authors have answered the previous comments from reviewers satisfactorily.

Experimental design

The authors have decided to investigate various groups of wound healing interventions. Each group then consists of individual studies.
It seems that the efficacy of the wound healing interventions is analysed as a group and not for each individual study.
If I am correct, then I feel that the authors should make this clear.

Validity of the findings

No comment

Additional comments

No comment

---

## Round 0.3 · accepted · Accept

Dear Author,

Congratulations! After your diligent work addressing the reviewers' comments, I am pleased to inform you that your manuscript has been accepted for publication in PeerJ. This version is more concise and formal, enhancing clarity and flow.

·

Basic reporting

Thank you for giving me another opportunity to review this work. This work much better read exploring the network analysis of DFU care. This discussion raised a new insight on how the DFU interventions related to each other in times. However, such intervention can not be implemented particularly in limited- resources countries. The recommendation should be made in that countries.

The writing style is fine and readable, no ambiguous words or sentences found during my read.

Experimental design

Since the study used a review desgin, the questions related to the experimental design is not relevant in this manuscript.

The method used in robust as the gold standard of review study.

Validity of the findings

The finding contributed to the recent need of DFU care in community or hospital settings. However, implementing of some intervention may challenging in some countries.

Additional comments
* * *
Reviewer 3 ·

Basic reporting

Satisfactory

Experimental design

Satisfactory

Validity of the findings

satisfactory